# The Effect of Task Performance and Partnership on Interpersonal Brain Synchrony during Cooperation

**DOI:** 10.3390/brainsci12050635

**Published:** 2022-05-11

**Authors:** Shujin Zhou, Yuxuan Zhang, Yiwen Fu, Lingling Wu, Xiaodie Li, Ningning Zhu, Dan Li, Mingming Zhang

**Affiliations:** 1Department of Psychology, Shanghai Normal University, Shanghai 100 Guilin Road, Xuhui District, Shanghai 200234, China; m13752093005@163.com (S.Z.); zhangyuxuan_psy@163.com (Y.Z.); fuyiwen1003@163.com (Y.F.); adan2526@163.com (L.W.); lixiaodie1025@163.com (X.L.); zningning347@163.com (N.Z.); lidan501@126.com (D.L.); 2 College of Marxism, Kashgar Vocational and Technical College, Kashgar, Xinjiang 844000, China

**Keywords:** interpersonal brain synchrony, cooperation, hyperscanning, cooperative partnership, task performance

## Abstract

Interpersonal brain synchrony (IBS) during cooperation has not been systematically investigated. To address this research gap, this study assessed neural synchrony during a cooperative jigsaw puzzle solving task using functional near-infrared spectroscopy (fNIRS)-based hyperscanning. IBS was measured for successful and failed tasks in 31 dyads in which the partners were familiar or unknown to each other. No significant difference in IBS was observed between the different types of cooperative partnership; however, stronger IBS within regions of the pars triangularis Broca’s area, right frontopolar cortex, and right temporoparietal junction was observed during task success. These results highlight the effect of better task performance on cooperative IBS for the first time and further extend understanding of the neural basis of cooperation.

## 1. Introduction

Cooperation is a critical form of social interaction wherein two or more people work together to achieve a common goal in a more or less organized way [1,2]. Cooperation is thought to have played an essential role in the evolution of human society [3], including social problems and problems about the human self [4]. Notably, cooperation facilitates complex problem solving [5], which is recognized as a feature that fundamentally distinguishes humankind from other species and is represented in the human brain [6]. An increasing amount of research has aimed to understand the nature of cooperation by measuring brain activity in two or more people during cooperative tasks [5,6,7,8,9]. Exploring the synchronization of two or more individuals’ neural systems, which is termed interpersonal brain synchrony (IBS), during a cooperative interaction can be assessed using functional near-infrared spectroscopy (fNIRS)-based hyperscanning technology [10,11].

Studies have consistently found that better cooperative interactions between individuals are associated with higher IBS [5,7,8]. Indeed, the accumulated hyperscanning research has reported that higher levels of IBS in the frontopolar cortex (FPC), orbitofrontal cortex (OFC), dorsolateral prefrontal cortex (DLPFC), superior frontal cortex (SFC), superior temporal gyrus (STG), and temporoparietal junction (TPJ) were associated with higher task performance during goal-oriented cooperative interactions [7,12,13,14,15,16,17,18]. For instance, IBS in the FPC, which has a role in social interaction [19], tended to increase with increasing task performance during cooperation [12]. In addition, strong IBS of the bilateral DLPFC has been observed for participants interacting cooperatively but not for participants performing a similar task independently [20]. It is notable that IBS is more likely to emerge during cooperative interactions in which the core components of interpersonal communication and shared attention are present [11]. For instance, greater IBS has been observed during face-to-face, but not back-to-back, conversations [14]. Overall, these findings suggest that IBS represents the quality of cooperation.

What causes us to fall in neural synchrony with others during cooperative interaction? Recent studies have revealed that IBS requires both partners to engage in the same social activities for common goals [7,18,21,22]. These studies have revealed some general factors that exist in interacting partners and social activities that likely influence IBS during cooperation. Factors related to the interacting partners include individual characteristics (i.e., age, gender) and their interpersonal relationship [23]. For example, Pan et al. [15] reported a significant increase in IBS in the r-SFC for dyads composed of lovers compared to male-female friends or stranger dyads. IBS in the TPJ and frontal cortex has been reported in studies of parent-child cooperative interactions [24]. In addition, greater IBS in the DLPFC and FPC was observed in parent–child pairs when compared to stranger-child pairs [22]. Thus, it appears that IBS is associated with an interacting partner who is considered significant. This may be because cooperation between partners with a stronger interpersonal relationship involves more interactive experiences and emotional connections. We exist in a world full of rich opportunities to cooperate with those considered important to us, such as lovers, peers, and parents [11,25]. Examples of interactive experiences include alignment of cognition [11], joint action [26], group problem solving [27], and affective interaction [28]. Hence, cooperative interactions involving close relationships may result in greater IBS than normal relationships.

Factors related to the social activities mainly include how the activity is performed and how well the activity is performed. Undoubtedly, the naturalistic setting of cooperative interactions and shared gaze during communication has been reported in many studies to strengthen IBS during cooperation [14]. On the other hand, IBS between different regions—including the PFC-TPJ and PFC-STG [13] and the DLPFC, FPC [22], and frontal cortex [12]—was significantly related to task performance. However, this association mainly refers to the predictive effect of IBS on task performance [7,17,22]. In comparison, few studies have systematically explored the influence of task performance on IBS. The effect of task performance on synchronization during cooperation is not an established phenomenon and thus warrants further research. Thus, it can be concluded that the major determinants of cooperative IBS are who we cooperate with and how well we perform together in the context of naturalistic cooperation.

Therefore, the primary goal of the present study is to determine the key trigger of cooperative IBS; is it who we cooperate with or how well we perform cooperatively? It is notable that dyads tend to show IBS when they perceive that their cooperative interaction is conducive to task performance [29]. Indeed, a growing body of research provides evidence of the positive correlation between IBS and task performance, particularly task success, regardless of who the individual cooperates with [30]. Gvirts and Perlmutter [11] proposed a framework of several factors that facilitate IBS during interaction. They believe that perceiving an interaction as significant is a necessary path to increase IBS. It is plausible that the effectiveness of cooperation is perceived more easily with better performance. Furthermore, achievable goals are more likely to result in better performance. As a result, compared to better partnership, better performance is more likely to induce IBS during cooperation.

The present study aimed to examine the roles of task performance and partnership in IBS to shed light on the nature of cooperative IBS. Of the tasks used in the field of cooperation research, the paradigms of solving problems and communicating seem more naturalistic, such as playing the game Jenga [16]. Do note, however, that task performance is the independent variable in present study. In order to clearly demonstrate the variable of task performance, the present study adopted a new task paradigm, a jigsaw puzzle, to investigate the nature of cooperative IBS using fNIRS-based hyperscanning. In our experiment, we manipulated the feasibility of the task by varying levels of difficulty: task success and task failure. In addition, familiar dyads and unknown dyads were generated to represent the variable of partnership. Here, task success is more likely to enhance cooperative IBS than better partnership. Previous hyperscanning studies of cooperation mainly report IBS in the frontal cortex and right TPJ [11,20]. Thus, we defined the region of interest (ROI) for the present study as the frontal and right TPJ areas. We hypothesized that: (1) there may be no difference in IBS between familiar dyads and unknown dyads during both task conditions; and (2) IBS will increase in the task success condition, but not the task failure condition.

## 2. Method

### 2.1. Participants

A total of 62 adults (31 dyads, mean age: 23 ± 1.50 years old, 37 females) participated in this study. Of the 31 dyads, 16 dyads were acquainted prior to the experiment and the remaining dyads were unknown to each other. Furthermore, the degree of closeness between acquainted dyads was measured by the inclusion of other in the self (IOS) scale [31] in order to ensure the reliability of study. In the IOS scale, each participant in the dyad selects the picture that best describes their relationship from a set of Venn-like diagrams, which representing different degrees of overlap between two circles, and there was no significant difference within the dyad (*t*_(15)_ = −1.142, *p* = 0.271, Cohen’s d = 0.363). Finally, the average score is used as the dyad closeness. There were 15 female–female dyads, 9 male-male dyads, and 7 male–female dyads. All participants were right-handed with normal or corrected vision. This study was approved by the local ethics committee of Shanghai Normal University and all participants provided written informed consent prior to participation. Participants were paid ¥30 for their involvement.

### 2.2. Experimental Tasks and Procedure

The experiment was carried out in a silent room. The two participants in the dyad sat side by side in front of a shared iPad (Figure 1A). The iPad, which has a 10.5-inch screen, remained at the same brightness throughout the experiment. The task used a free puzzle application named Jigsaw Puzzle (developed by Guihang Xu) that was already open on the iPad. After collecting their demographic data, the experimental task was explained in detail to each dyad. The procedure consisted of one task that could be successfully completed (time unlimited) and one task that could not be successfully completed (5 min) [32], with rest times of 30 s before starting the procedure, between tasks, and after finishing the procedure (Figure 1B). For each condition (task success and task failure), the dyad was asked to solve one jigsaw puzzle problem together under different time requirements. Specifically, the goal image of the jigsaw puzzle was a cat (Figure 1C), which was comprised of 35 pieces in the task success condition and 140 pieces in the task failure condition. All pieces were presented on the right side of the screen and could be freely dragged by touch. During the task, the two participants were instructed to take turns placing one piece. They were told that they could discuss the strategy with each other and agree on a specific step. The allotted time was evaluated by all participants with a similar image before the experiment in order to validate the solving time allocation. All participants failed to solve the task failure condition puzzle in 5 min.

### 2.3. fNIRS Data Acquisition

Continuous wave fNIRS (NIRScout 24 × 24, NIRx Medizintechnik GmbH, Germany) was used to assess cortical hemodynamic activity using two wavelengths (760 and 850 nm). The sampling frequency was 7.8125 Hz. Forty-eight optodes (24 sources, 24 detectors) were divided between the two participants of each dyad, forming 26 channels per participant (8 sources, 12 detectors) covering the frontal and right TPJ regions (Figure 2). NIRS optodes were positioned on the subject’s head using an NIRS cap according to the international 10/20 system. Plastic supports were placed between each source/detector pair that constituted the 3 cm channel length.

We assessed the anatomical positions of optodes and channels (CHs) on a standardized 3D head with Nz (nasion), Iz (Inion), AL (left preauricular point), AR (right preauricular point), and Cz (central zero) as referential points. Then, the NIRS_SPM software (http://www.nitrc.org/projects/nirs_spm/, accessed on 8 December 2021) was used to estimate the Montreal Neurological Institute (MNI) coordinates of optodes and further obtain the probabilistic cortical localization of all 26 channels (Table 1) [33,34]. 

Regions of interest (ROI) were created by the fNIRS optodes’ location decider (fOLD) [35] and NIRS-SPM toolboxes in MATLAB. ROIs were grouped as follows: (1) left frontal lobe: CH 2, 3, 4, 5, 6, 7, 8, 9, 10, and 11; (2) right frontal lobe: 13, 14, 15, 16, 17, 18, 19, 20, and 21; (3) left temporal lobe: 1; (4) right temporal lobe: 22, 23, 24, and 26; and (5) postcentral gyrus: 25.

### 2.4. Behavior Data Analysis

The puzzle application recorded the completion time spent for dyad in each task success condition. Completion time referred to the time length required for the dyad from the beginning of cooperation to the completion of the target image. Dyadic task performance in the task failure condition was indexed by the degree of completion. The degree of completion during the limited time was rated by two graduate students on a 7-point Likert scale (1 = low degree of completion, 7 = high degree of completion) from the video recordings and averaged between the two raters. The inter-rater reliability of the degree of completion (α = 0.81) was satisfactory. Finally, through SPSS Statistics 19, we conducted independent sample t-tests on the completion time and the degree of completion between dyad compositions respectively.

### 2.5. fNIRS Data Analysis

Data were preprocessed using the HOMER2 package implemented in MATLAB. First, the raw intensity data were converted into optical density (OD) data with the *hmrIntensity2OD.m* function. Next, the optical density data were corrected for motion artifacts using the *hmrMotionArtifact.m* function with the following parameters: tMotion = 0.5, tMask = 1, STDEVthresh = 10, and AMPthresh = 1. PCA filtering was then applied for motion artifact removal using *hmrMotionCorrectPCA.m* with the parameter nSV set to 0.8. Lastly, the oxygenated hemoglobin (HBO) and deoxygenated hemoglobin (HBR) concentrations were calculated from the filtered optical density data using the *hmrOD2Conc.m* function. We focused only on the HbO time series because the HbO signal has been shown to be more sensitive to changes in cerebral blood flow [7,14].

Inter-subject coherence analysis was performed for each effective channel using the wavelet transform coherence (WTC) package developed by Grinsted et al. [37] and Chang and Glover [38] and custom MATLAB code (MathWorks). WTC was used to assess the relationship between the HbO time series for each dyad (for more details, see Grinsted et al. [37]). Based on the WTC analyses of the two-time series generated by each dyad, we focused on the frequency band between 6.9 s and 7.3 s that was more sensitive to our task (Figure 3). This frequency band did not include high- and low-frequency noise, such as physiological noises related to Mayer waves (0.1 Hz), respiration (~0.2–0.3 Hz) and cardiac pulsation (about 1 Hz) [39,40]. The average coherence in this band for each task condition was calculated by subtracting the average coherence during the rest session from that during the task session. Finally, the averaged coherence values were converted to Fisher *z*-statistics [7,38]:(1)$$Fz=\frac{1}{2}\ln\left(\frac{1+C}{1-C}\right)$$

Subsequently, paired-sample *t*-tests were performed using data for the task success and task failure conditions to determine the difference in IBS. To test whether IBS differed between the task success and task failure conditions in different dyad composition types, we performed 2 (condition: task success vs. task failure) ×2 (dyad composition: familiar dyads vs. unknown dyads) mixed-design analyses of variance (ANOVAs), using condition as a within-dyad factor and dyad composition as a between-dyad factor.

## 3. Results

### 3.1. Behavioral Data 

The differences of the completion time and the degree of completion between the two dyad compositions were examined by independent sample t-tests. The results showed that there was no significant difference of the completion time [*t*_(29)_ = −1.64, *p* = 0.11, Cohen’s d = 0.59] and the degree of completion [*t*_(29)_ = 1.43, *p* = 0.16, Cohen’s d = 0.52] between the two dyad compositions.

### 3.2. Interpersonal Brain Synchronization (IBS)

We first conducted a series of independent t-tests to investigate the difference in IBS between the two dyad compositions under the task success condition and the task failure condition, respectively. With respect to it, no channel with significant IBS was found.

Subsequently, two-way mixed design ANOVA with dyad composition as the between-subject factor and task condition as the within-subject factor was performed on the IBS. The ANOVA results revealed a significant main effect of task condition (*F*_(1, 29)_ = 7.651, *p* = 0.01, η^2^ = 0.209). However, the main effect of dyad composition and the interaction effect of condition and dyad composition were not significant (*F*_(1, 29)_ = 0.301, *p* = 0.588, η^2^ = 0.01; *F*_(1, 29)_ = 0.037, *p* = 0.695, η^2^ = 0.005) (Figure 4B).

Further analysis for the main effect of the task condition using a paired-sample *t*-test, compared with the task failure condition revealed higher IBS in the task success condition at channel 3 (*t*_(30)_ = 2.522, *p* = 0.017, Cohen’s d = 0.699) and channel 21 (*t*_(30)_ = 3.067, *p* = 0.005, Cohen’s d = 0.827), which are located in the pars triangularis Broca’s area (Brodmann area, BA 45). The coherence increase was significant in channel 14, which is located in the FPC (BA 10; *t*_(30)_ = 2.456, *p* = 0.020, Cohen’s d = 0.648). IBS was also significant in the right TPJ at channel 26 (*t*_(30)_ = 2.180, *p* = 0.037, Cohen’s d = 0.534). Figure 4A shows the t-values of the HbO signal difference between the task success and task failure conditions visualized on a brain cortex template using the Xjview toolbox (http://www.alivelearn.net/xjview, accessed on 26 December 2021) and BrainNet Viewer toolbox [41].

### 3.3. The IBS-Behavior Relation 

Bivariate Pearson correlations were performed on the regions of significant IBS (e.g., BA 10, BA 45) and behavior indices. Results showed that interbrain coherence in regions of rDLPFC (channel 19: *r* = 0.50, *p* < 0.01) and BA 45 (channel 20: *r* = 0.37, *p* < 0.05) was positively correlated with completion time. In contrast, measured interbrain coherence in the frontal regions (channel 13: *r*= −0.51, *p* < 0.01) was negatively correlated with the degree of completion. Additionally, the degree of closeness was negatively associated with IBS in regions of rDLPFC (channel 6: *r* = −0.51, *p* < 0.05; channel 19: *r* = −0.60, *p* < 0.05) and BA 45 (channel 3: *r* = −0.58, *p* < 0.05) for the task success condition.

## 4. Discussion

Although the factors that facilitate IBS have been previously studied, the factors triggering cooperative IBS have yet to be studied. To address this research gap, the present study sought to determine the key trigger of cooperative IBS by comparing cooperative performance and partnership during a naturalistic puzzle-solving paradigm using the fNIRS-based hyperscanning technique. The results showed that task performance had a significant influence on cooperative IBS. Overall, these findings provide new and fundamental information that better task performance is more likely to be the key trigger of cooperative IBS than a more interactive partnership.

Above all, the behavioral results showed that there was no significant difference between the two dyad compositions for the different performance indices under different conditions. Moreover, this is combined with the fNIRS results that showed no significant difference between the two dyad compositions. The results indicated that there was no significant effect of partnership to IBS. This finding appears to be at odds with evidence of increased IBS with better partnership. A likely explanation for this result may be found in the theory of distraction/conflict [42,43], which argues that positive peer relationships may inhibit optimal performance. Specifically, more positive partnerships are associated with greater attentional temptation, distracting the attention needed in the tasks, and thus resulting lower task performance. As the results demonstrated with the negative relationship between the degree of closeness and IBS in the rDLPFC and BA 45 regions, the closer the friends were, the worse the task fluency was.

The IBS results indicated that both familiar dyads and unknown dyads were capable of a solving a puzzle cooperatively in the task success condition and showed increased IBS at the pars triangularis Broca’s area (BA 45), FPC (BA 10), and rTPJ. The pars triangularis Broca’s area, a sub-region of the inferior frontal gyrus (IFG), is crucially involved in language processing [44,45]. Clearly, the dyads need to talk to each other in order to solve the puzzle. Importantly, BA 45 is not simply a speech production region of the brain, it is a sophisticated region involved in action understanding, verbal working memory, and verbal fluency [46]. Syntactic verbal fluency is sensitive to the difficulty of the selection in verbal working memory [46]. Since the jigsaw puzzle in the task success condition was easier, the dyads had enough time to exchange observed information and engage in more analytical cognitive processing, and thus made verbal suggestions and strategies more fluently than in the task failure condition. This interpretation is consistent with previous evidence that the IFG is associated with analytic processing of semantic information [47] and responses to observations of actions [48]. Accordingly, it is plausible that BA 45 may reflect the functions of “mirror neurons”, including verbal fluency and action understanding [46]. Therefore, it is conceivable that channels in BA 45 were positively correlated with performance under task completion conditions. In general, it can be concluded that BA 45 of the IFG has the adaptive function of promoting phonetics and action sharing during cooperative interactions [49].

As predicted, strong IBS was observed at FPC (BA 10) and rTPJ during the task success condition when compared to the task failure condition. Previous studies have demonstrated that BA10 and rTPJ are mentalizing system regions crucial for tasks involving goal-oriented social interaction and shared intentionality [16]. In contrast, the BA 10 has been consistently shown to be engaged in coordination-related cognitive control and goal maintenance [50,51]. The findings in BA 10 are also in line with that of Li et al. [52], who suggested that BA 10 may play a vital role in complex turn-based interactive movement interactions with common goals. The rTPJ plays an essential role in inferring others’ intentions [53], desires, and beliefs [54] in order to build a correct mental model. During our puzzle task, the individuals in the dyad made decisions about the next step (i.e., which puzzle piece to drag and where to put it) based on their partner’s suggestions and responses (i.e., “we can start with the four corners of the puzzle”) in order to complete the task in 5 min. As noted above, compared with the failed puzzle, there are more cognitive resources available in the task success condition for dyads to maintain the task goal and attention in turn-based cooperation and generate further strategies. In the same vein, the task under the failure condition required more cognitive resources, even with a small increase in the degree of completion, resulting in a lower IBS in the prefrontal region, responsible for executive functions [55]. Overall, our results suggest that the observed coherence in BA 10 and rTPJ might relate to shared attention and goal maintenance in turn-based cooperative interaction when the goal is more feasible.

Notably, task performance has a significant influence on cooperative IBS. According to the social interdependence theory, better cooperation exists only when the dyads mutually perceive that they can reach their goals [42]. With that, achievable goals and mutual attention are the essential elements of higher cooperative IBS. The task success condition in the current study, the achievable jigsaw, is more likely to result in spontaneous attention. Indeed, mutual attention serves as a basis for social interactions [56]. In detail, perceiving an interaction as significant, which requires mutual attention, serves as a synchronization trigger, resulting in neural and behavioral alignment with partners [11]. In actuality, cooperation interaction with better cooperative performance is more likely to be considered profitable. Moreover, greater IBS within the FPC and rTPJ regions was associated with shared attention only for the task success condition. This result supports the roles of achievable goals and mutual attention in facilitating more attunement, and greater allocation of attention.

In summary, the present study aims to understand the nature of IBS during cooperation by comparing the effects between cooperative performance and cooperative partnership. Our findings have several implications. Firstly, compared to better partnership, better task performance has a greater effect on triggering cooperative IBS. In contrast to previous research that focused on the predictive effect of IBS on cooperative performance, this study emphasizes the predictive effect of cooperative performance on IBS; in other words, this study provides evidence for the bidirectional link between them. Secondly, increased IBS within the pars triangularis Broca’s area, FPC, and rTPJ represents the cognitive process of turn-based cooperation: sharing intentionality and jointly making strategic decisions through verbal communication [22,53]. Lastly, as outlined by Gvirts and Perlmutter [11], three aspects of interaction facilitate IBS: (i) type of social activity (i.e., interactive vs. non-interactive in Liu et al. [16]), (ii) the setting of the interaction (i.e., face to face vs. back to back in Jiang et al. [14]), and (iii) the nature of the interacting partners (i.e., same-sex dyad vs. opposite-sex dyad in Cheng et al. [12]). In the same vein as this framework, we propose three factors that facilitate IBS during cooperation: (i) the setting of cooperation (i.e., face-to-face cooperative interaction with shared gaze vs. back-to-back cooperative interaction without eye contact), (ii) the nature of the cooperating partners (i.e., goal-oriented groups vs. groups without common goals), and (iii) cooperative performance (i.e., goals accomplished vs. goals failed). 

Although the present study is the first to examine the influence of task performance on IBS during cooperation within multiple regions of the frontal and temporal cortices, important aspects of better task performance are likely also associated with neural reward systems. However, as the relevant brain structures are mostly subcortical, we were unable to target them. These subcortical structures can be investigated using functional magnetic resonance imaging (fMRI)-based techniques in the future. Additionally, given the experimental design where one puzzle must be completed and the other cannot be completed, it unfortunately leads to inconsistent performance units between the two conditions. Moreover, given that the negative emotions induced by failed tasks have a greater impact on some individuals [57], we did not balance the sequence of task success and task failure conditions. In addition, the gender of the dyads was not considered in the present study due to the small sample size. As opposite gender dyads may show different levels of coherence, future studies should expand the sample size to clarify the effect of gender. Finally, future studies could detail the framework of IBS during cooperation by exploring the impact of the three aspects proposed above. 

## 5. Conclusions

The hyperscanning approach presented in this study sheds light on the factors triggering IBS during naturalistic interpersonal cooperative interactions. Our results indicated a significant effect only for better task performance. Increased IBS in the pars triangularis Broca’s area, FPC, and rTPJ were observed for successful tasks, but not for failed tasks. Thus, better task performance may be the key trigger of cooperative IBS. This finding allows us to develop a better understanding of the nature of naturalistic cooperation as seen through the lens of IBS.

## Figures and Tables

**Figure 1 brainsci-12-00635-f001:**
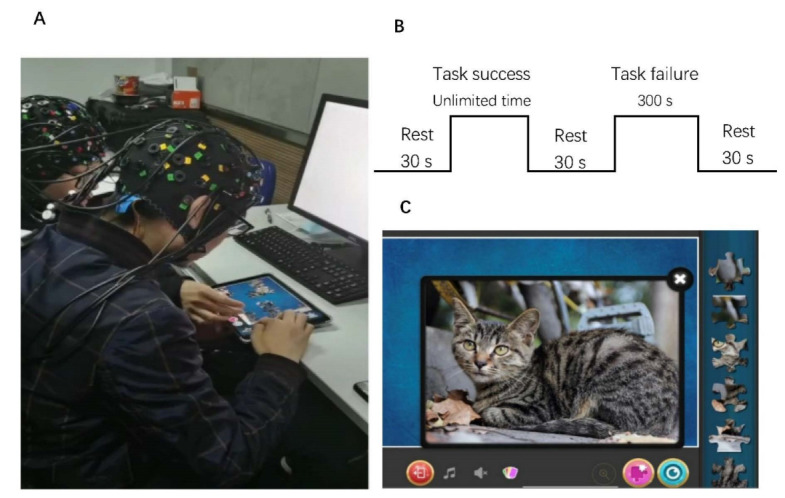
Experimental design. (**A**) Participants sat side by side at a square table in front of a shared iPad. (**B**) Hyperscanning design: rest 1 (30 s), task success condition (unlimited time), rest 2 (30 s), task failure condition (300 s), and rest 3 (30 s). (**C**) The goal image of the jigsaw puzzle.

**Figure 2 brainsci-12-00635-f002:**
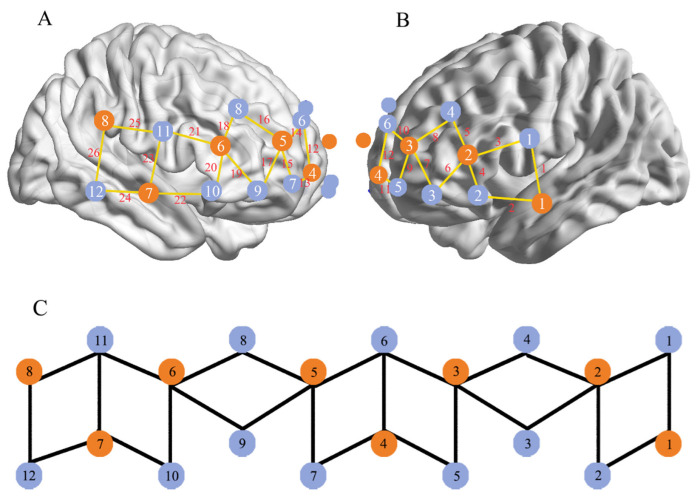
Map of the Nirscout fNIRS channels’ and probes’ locations in the right (**A**) and left (**B**) hemispheres. Legend of mapping colors: sources are in orange, detectors are in blue, and channels are in yellow. (**C**) represents the full montage of all channels’ combinations.

**Figure 3 brainsci-12-00635-f003:**
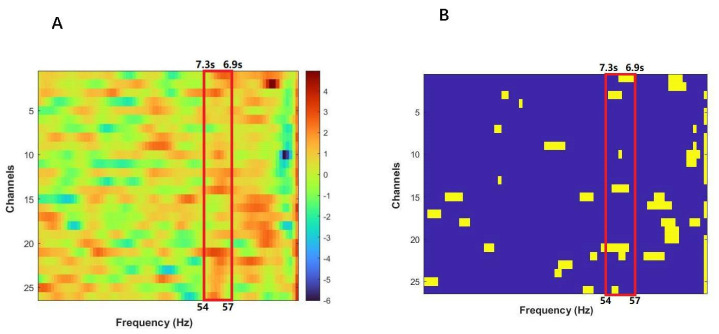
(**A**) T-value map resulting from a series of paired-sample t-tests conducted on task-related WTC which was defined as increased coherence in the difference between the task success and task failure. (**B**) Channels with *p* < 0.05 are marked by yellow blocks. The red border covers the frequency band ranging from 54 to 57.

**Figure 4 brainsci-12-00635-f004:**
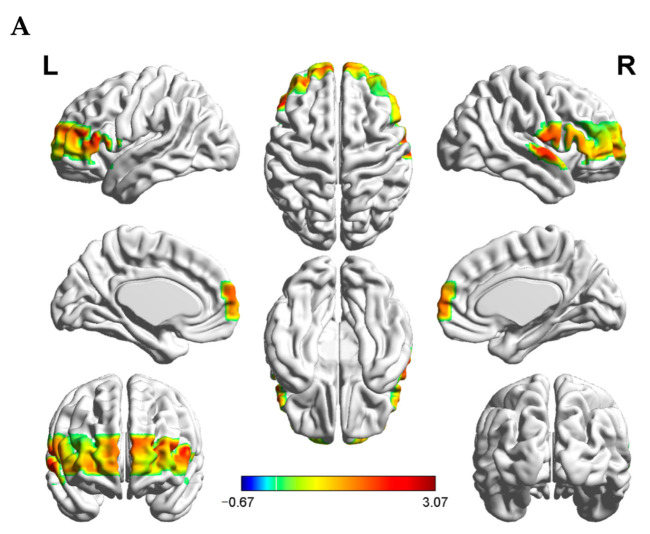
(**A**) T-maps of the cortical activation difference between the task success and task failure conditions. (**B**). The amplitude of coherence in the FPC (channel 14) for task condition and dyad composition. *, *p* ≤ 0.05. Error bars indicate the standard error of the mean. The *Y*-axis represents the coherence value at channel 14.

**Table 1 brainsci-12-00635-t001:** The MNI coordinates and probabilistic cortical localization of all 26 channels.

Channels	MNI Coordinates			Brodmann’s Areas	Percentage of Overlap
	x	y	z		
1 FT7-FC5	−62	8	5	48—Retrosubicular area 6—Pre-Motor and Supplementary Motor Cortex	0.620 0.295
2 FT7-F7	−55	17	−13	38—Temporopolar area	0.986
3 F5-FC5	−57	28	16	45—pars triangularis Broca’s area 44—pars opercularis, part of Broca’s area	0.845 0.155
4 F5-F7	−54	40	0	45—pars triangularis Broca’s area 46—Dorsolateral prefrontal cortex	0.717 0.257
5 F5-F3	−47	45	24	45—pars triangularis Broca’s area 46—Dorsolateral prefrontal cortex	0.751 0.249
6 F5-AF7	−48	51	0	46—Dorsolateral prefrontal cortex	0.929
7 AF3-AF7	−36	64	3	10—Frontopolar area 11—Orbitofrontal area	0.820 0.132
8 AF3-F3	−33	57	26	46—Dorsolateral prefrontal cortex	0.966
9 AF3-Fp1	−24	70	5	10—Frontopolar area 11—Orbitofrontal area	0.698 0.302
10 AF3-Afz	−13	68	24	10—Frontopolar area	0.997
11 Fpz-Fp1	−12	73	−4	11—Orbitofrontal area 10—Frontopolar area	0.505 0.495
12 Fpz-Afz	2	68	13	10—Frontopolar area	1
13 Fpz-Fp2	14	73	−4	11—Orbitofrontal area 10—Frontopolar area	0.516 0.484
14 AF4-Afz	16	69	24	10—Frontopolar area	1
15 AF4-Fp2	27	70	6	10—Frontopolar area 11—Orbitofrontal area	0.721 0.279
16 AF4-F4	36	57	27	46—Dorsolateral prefrontal cortex	0.956
17 AF4-AF8	40	64	4	10—Frontopolar area 46—Dorsolateral prefrontal cortex	0.798 0.107
18 F6-F4	49	44	25	45—pars triangularis Broca’s area 46—Dorsolateral prefrontal cortex	0.809 0.191
19 F6-AF8	50	51	1	46—Dorsolateral prefrontal cortex	0.906
20 F6-F8	57	38	1	45—pars triangularis Broca’s area 46—Dorsolateral prefrontal cortex	0.803 0.188
21 F6-FC6	60	27	18	45—pars triangularis Broca’s area 44—pars opercularis, part of Broca’s area	0.825 0.175
22 FT8-F8	59	15	−13	38—Temporopolar area 21—Middle Temporal gyrus	0.881 0.119
23 FT8-FC6	64	7	6	48—Retrosubicular area 6—Pre-Motor and Supplementary Motor Cortex	0.627 0.300
24 FT8-T8	71	−10	−12	21—Middle Temporal gyrus	0.990
25 C6-FC6	69	−5	25	43—Subcentral area	0.949
26 C6-T8	73	−23	7	22—Superior Temporal Gyrus 21—Middle Temporal gyrus	0.717 0.283

The MNI coordinates were transformed to Talairach space [36] and looked up in a brain atlas. A NIRS channel may cover several brain regions and the percentages of overlap should sum up to 1. Here, we only report the brain regions cover more than 10% of the channel path.

## Data Availability

Code used in this analysis and derived data that support the conclusions of this study are available upon direct request to the corresponding author.

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
