# Peer review of "The Effect of Task Performance and Partnership on Interpersonal Brain Synchrony during Cooperation"

_brainsci, 2022, doi:10.3390/brainsci12050635_

Round 1
Reviewer 1 Report
Minor points:
1-In the section dedicated to “fNIRS data acquisition”, the following information should be added to allow for reproducibility.
-Please specify the fNIRS montage.
-Please specify the correspondence between the optodes positions and the EEG positions.
-Please specify how you determined the ROIs, if you adopted specific atlases or a software (such as fOLD) to determine the correspondence between channels and Broadmann areas.
-Please specify how you grouped the channels to create the ROIs.
2-In the section dedicated to “fNIRS data analysis” please specify explicitly how you reduced the influence of Mayer’s wave.
3- To allow for reproducibility, please specify how you converted the coherence values into Fisher z-statistics.
4-In figure 4b please specify the correspondence of the values on the y axis. Please specify the significant effects with standard indication, for example, the asterisk.
5- I suggest providing at least a brief potential explanation to the lack of significant effects for partnership, after the following sentence:
“What’s more, combined with the fNIRS results of no significant difference between the two dyad compositions. These findings indicated that there is no significant effect of partnership to the IBS.”
6-Page 9, second paragraph, line 4: please check the punctuation.
7-With reference to this proposal: “In the same vein as this framework, we propose three factors that facilitate IBS during cooperation: (i) the setting of cooperation, (ii) the nature of the cooperating partners, and (iii) cooperative performance.” I suggest explaining in parentheses what you mean by each of these factors and how they differ from the three aspects of interaction that facilitate IBS during cooperation.
Reviewer 2 Report
The Authors investigate Interpersonal brain synchrony (IBS) by fNIRS; they reveal a relationship of IBS with task success in Broca’s area, right dorsolateral prefrontal cortex, and right temporoparietal junction, but they find no relation of IBS with partner familiarity.
The subject of the work is interesting, however, I see some major points that need to be addressed:
1) A major issue is the localization of brain activity, especially DLPFC: the Authors should present and discuss relevant references, such as De Witte et al., 2018, according to which DLPFC is not correctly identified with the 10-20 syste. Therefore, IBS activity the Authors claims to be recorded in DLPFC appears as not actually located in DLPFC (see De Witte et al., 2018, Figs. 1-2). If this is the case, consequently, the Discussion about DLPFC should be either revised according to the correct localization, or altogether deleted.
2) Another relevant issue is the degree of closeness between the dyads: the Authors make a very generic statemente ("16 dyads were acquainted prior to the experiment"), but this is insufficient to warrant the statement "These findings indicated that there is no significant effect of partnership to the IBS", as the degree of acquaintance, closeness, and partnership can be extremely variable
3) Reference list should be expanded with recent works relevant to IBS, see, e.g.,: Anzolin et al. 2020 Annu Int Conf IEEE Eng Med Biol Soc, and/or Astolfi et al. 2020 Neuroimage.
Minor:
- References are quoted in full in the text, instead of numbers
- Abstract and Discussion: "regions of the pars triangularis Broca’s area": "regions of the pars triangularis of Broca’s area"
- Sect. 2.4 and 3.1: "degree of completion"
